# LRRK2 is a negative regulator of *Mycobacterium tuberculosis* phagosome maturation in macrophages

Anetta Härtlova[1,2,†], Susanne Herbst[3,4,†], Julien Peltier[1,2], Angela Rodgers[3], Orsolya Bilkei-Gorzo[1], Antony Fearns[3], Brian D Dill[1], Heyne Lee[5], Rowan Flynn[5], Sally A Cowley[5], Paul Davies[1], Patrick A Lewis[6,7], Ian G Ganley[1], Jennifer Martinez[8], Dario R Alessi[1], Alastair D Reith[9], Matthias Trost[1,2,*] & Maximiliano G Gutierrez[3,**]

## Abstract

Mutations in the leucine-rich repeat kinase 2 (LRRK2) are associated with Parkinson's disease, chronic inflammation and mycobacterial infections. Although there is evidence supporting the idea that LRRK2 has an immune function, the cellular function of this kinase is still largely unknown. By using genetic, pharmacological and proteomics approaches, we show that LRRK2 kinase activity negatively regulates phagosome maturation via the recruitment of the Class III phosphatidylinositol-3 kinase complex and Rubicon to the phagosome in macrophages. Moreover, inhibition of LRRK2 kinase activity in mouse and human macrophages enhanced *Mycobacterium tuberculosis* phagosome maturation and mycobacterial control independently of autophagy. *In vivo*, LRRK2 deficiency in mice resulted in a significant decrease in *M. tuberculosis* burdens early during the infection. Collectively, our findings provide a molecular mechanism explaining genetic evidence linking LRRK2 to mycobacterial diseases and establish an LRRK2-dependent cellular pathway that controls *M. tuberculosis* replication by regulating phagosome maturation.

**Keywords** LRRK2; Parkinson's disease; phagosome; Rubicon; tuberculosis
**Subject Categories** Membrane & Intracellular Transport; Microbiology, Virology & Host Pathogen Interaction
**The EMBO Journal** (2018) 37: e98694

## Introduction

Tuberculosis (TB) is an infectious disease caused by the intracellular pathogen *Mycobacterium tuberculosis* (Mtb), which is characterised by chronic inflammatory responses (Russell, 2011; Kaufmann & Dorhoi, 2013). TB is one of the most pernicious infectious diseases borne by mankind with an estimated 1.6 million deaths in 2016 (WHO, 2017). Mtb infects mostly macrophages and within these cells establishes a replicative niche by subverting the host cell and the normal process of phagosome maturation. This cellular pathway, that represents the cornerstone of the innate immune system, targets Mtb to phagolysosomes where they are eventually eliminated (Levin *et al*, 2016).

Mutations in the multi-domain leucine-rich repeat kinase 2 (LRRK2) are associated with inflammatory diseases such as Crohn's disease and ulcerative colitis (Zhang *et al*, 2009; Franke *et al*, 2010; Liu *et al*, 2011; Umeno *et al*, 2011; Marcinek *et al*, 2013). Additionally, genomewide association studies have implicated LRRK2 in mycobacterial immunopathology and identified LRRK2 as a risk factor for inflammatory responses in leprosy, an infection by the intracellular pathogen *Mycobacterium leprae* (Zhang *et al*, 2009; Wang *et al*, 2015; Fava *et al*, 2016) as well as Mtb (Wang *et al*, 2018). However, the implicated cellular pathways linking LRRK2 function and immunity are poorly characterised.

Mutations in LRRK2 represent a genetic risk associated with dominantly inherited and sporadic Parkinson's disease (PD; Funayama *et al*, 2002; Paisan-Ruiz *et al*, 2004; Zimprich *et al*, 2004; Ross *et al*, 2011). In PD, mutations in LRRK2 are distributed over

1 MRC Protein Phosphorylation and Ubiquitylation Unit, University of Dundee, Dundee, UK
2 Newcastle University, Newcastle-upon-Tyne, UK
3 Host-Pathogen Interactions in Tuberculosis Laboratory, The Francis Crick Institute, London, UK
4 Crick-GSK Biomedical LinkLabs, GlaxoSmithKline Pharmaceuticals R&D, Stevenage, UK
5 Sir William Dunn School of Pathology, University of Oxford, Oxford, UK
6 University of Reading, Reading, UK
7 UCL Institute of Neurology, Queen Square, London, UK
8 NIEHS, Research Triangle Park, NC, USA
9 Neurodegeneration Discovery Performance Unit, RD Neurosciences, GlaxoSmithKline Pharmaceuticals R&D, Stevenage, UK
 *Corresponding author. Tel: +44 191 2087009; E-mail: matthias.trost@ncl.ac.uk
 **Corresponding author. Tel: +44 203 7961460; E-mail: max.g@crick.ac.uk
 †These authors contributed equally to this work

the kinase and Ras of complex proteins (ROC) domains. Most of these mutations, including the most frequent mutation G2019S, are characterised by enhanced kinase activity (West *et al*, 2005). Despite evidence showing a strong association between LRRK2 activities and the regulation of intracellular trafficking and lysosomal degradation pathways (Alegre-Abarrategui *et al*, 2009; Tong *et al*, 2010; MacLeod *et al*, 2013; Manzoni *et al*, 2013; Orenstein *et al*, 2013; Steger *et al*, 2016), how pathogenic mutations regulate LRRK2 function remains elusive (Cookson, 2010).

Compelling evidence supports the idea that LRRK2 may have an important immune function (Dzamko & Halliday, 2012; Greggio *et al*, 2012). In fact, myeloid cells such as monocytes and macrophages express LRRK2 at high levels (Gardet *et al*, 2010; Hakimi *et al*, 2011) and several immune stimuli induce LRRK2 expression (Gardet *et al*, 2010; Hakimi *et al*, 2011). In this work, we examined the function of LRRK2 in macrophages and show that LRRK2 negatively regulates phagosome maturation and that this contributes to mycobacterial replication and impaired innate immune responses. Using several independent experimental approaches, we show that phagosomal function is regulated by a LRRK2 kinase-dependent recruitment of the Class III phosphatidylinositol-3 kinase (PI3K) complex and its negative regulator Rubicon (RUN domain protein as Beclin-1 interacting and cysteine-rich containing). Strikingly, macrophages lacking LRRK2 or treated with an inhibitor of LRRK2 kinase activity showed more efficient control of Mtb replication by macrophages and an enhanced early immune response in LRRK2 KO mice. This study provides a cellular function underlying human genetic studies linking LRRK2 to mycobacterial infections and reveals an unexpected function for LRRK2 in macrophages and infectious diseases control.

# Results

## Loss of LRRK2 activity targets Mtb to phagolysosomes and limits Mtb replication

Given that LRRK2 is highly expressed in macrophages and several human genetic studies linked LRRK2 and mycobacterial diseases (Zhang *et al*, 2009; Wang *et al*, 2015, 2018; Fava *et al*, 2016), we investigated the effect of LRRK2 on Mtb infection using bone marrow-derived mouse macrophages (BMDMs) from LRRK2 KO mice. LRRK2 KO macrophages were able to control Mtb replication significantly better (Fig 1A). Confirming this result in human cells, LRRK2 KO human-induced pluripotent stem cell-derived macrophages (iPSDM) also significantly restricted Mtb replication (Fig 1A). Notably, treatment of both BMDM and iPSDM with the LRRK2 kinase inhibitor GSK2578215A (Fig EV1A) significantly restricted Mtb replication (Fig 1B), indicating that inhibition of the LRRK2 kinase activity enhanced Mtb control by macrophages. The improved control of Mtb was not due to enhanced macrophage toxicity in LRRK2 KO or GSK2578215A-treated macrophages (Fig EV1B and C). Interestingly, the secretion of the pro-inflammatory cytokines TNF-α and IL-6 was not altered in LRRK2 KO or GSK2578215A-treated BMDMs infected with Mtb (Fig 1C and D), whilst IL-10 secretion was significantly down-regulated (Fig 1C and D). In agreement with an enhanced Mtb control in LRRK2 KO macrophages, the percentage of Mtb positive for the late endocytic/lysosomal marker

LAMP-1 was at least twofold higher in LRRK2 KO than in WT macrophages (Fig 1E and F). Moreover, the fraction of Mtb in proteolytic phagosomes was higher in LRRK2 KO macrophages as measured by the activity of the lysosomal enzyme cathepsin L (Fig 1G and H). Consistently, GSK2578215A treatment also resulted in a remarkable increase of co-localisation between LAMP-1 as well as active cathepsin L with Mtb (Fig 1I–L). Similar results were obtained in the mouse macrophage cell line RAW264.7 (Fig EV1D–F). In contrast, macrophages harbouring the LRRK2 gain-of-function mutation G2019S were more susceptible to Mtb replication (Fig 1M) and showed a reduction in lysosomal targeting of Mtb to phagolysosomes as measured by LAMP-1 recruitment (Fig 1N and O). An image-based approach (Schnettger *et al*, 2017) confirmed reduced Mtb growth in LRRK2 KO macrophages and enhanced growth in G2019S KI macrophages (Fig EV1G and H). Moreover, IFN-γ activation and control of Mtb was not synergistic (Fig EV1I) and enhanced Mtb phagosome maturation was not due to a general defect in late endosomal morphology (LAMP-1) or CtsL activity (Fig EV2). In contrast to LRRK2 KO BMDM, G2019S KI BMDM showed reduced secretion of both TNF-α and IL-6 with increased secretion of IL-10 (Fig 1O). Altogether, these results indicate that LRRK2 and its kinase activity affect not only Mtb replication in both human and mouse macrophages by regulating phagosome maturation but also cytokine responses.

## LRRK2 does not alter autophagic targeting of *Mycobacterium tuberculosis*

LRRK2 inhibition has been described to induce autophagy (Manzoni *et al*, 2013), a process which targets Mtb to phagolysosomes (Gutierrez *et al*, 2004). Therefore, we investigated whether LRRK2 KO resulted in Mtb growth restriction via the induction of autophagy. As reported previously, LRRK2 kinase inhibition induced LC3BII accumulation and reduced p62 levels over time (Fig EV3A and B), indicative for autophagy induction. Additionally, we saw a minor but reproducible increase in LC3BII levels in LRRK2 KO macrophages (Fig EV3B and C). However, this increase seemed to be independent of autophagic flux as there were no differences in LC3BII accumulation after Bafilomycin A1 treatment between WT BMDMs, BMDMs treated with GSK2578215A or LRRK2 KO BMDMs (Fig 2A and B). In agreement with these data, there were no differences between WT and LRRK2 KO BMDM in LC3B processing and p62 degradation after induction of autophagy with IFN-γ or nutrient starvation (Fig EV3D and E). Mtb itself is able to block autophagic flux in macrophages. Concordantly, we detected LC3BII and p62 accumulation in infected WT macrophages over time. However, we did not observe any differences between WT and LRRK2 KO macrophages, indicating that LRRK2 KO is not able to overcome the Mtb-induced block in autophagic flux (Fig 2C and D). Given that we observed increased autophagy in infected macrophages, we analysed the targeting of Mtb to LC3B- or p62-positive compartments. There was targeting of Mtb to LC3B- and p62-positive compartments; however, there were no significant differences in the number of Mtb positive for LC3B and p62 across WT, LRRK2 KO and G2019S KI macrophages (Fig 2E and H). We concluded that although LRRK2 function alters autophagy, it does not regulate Mtb targeting to autophagosomes in macrophages.

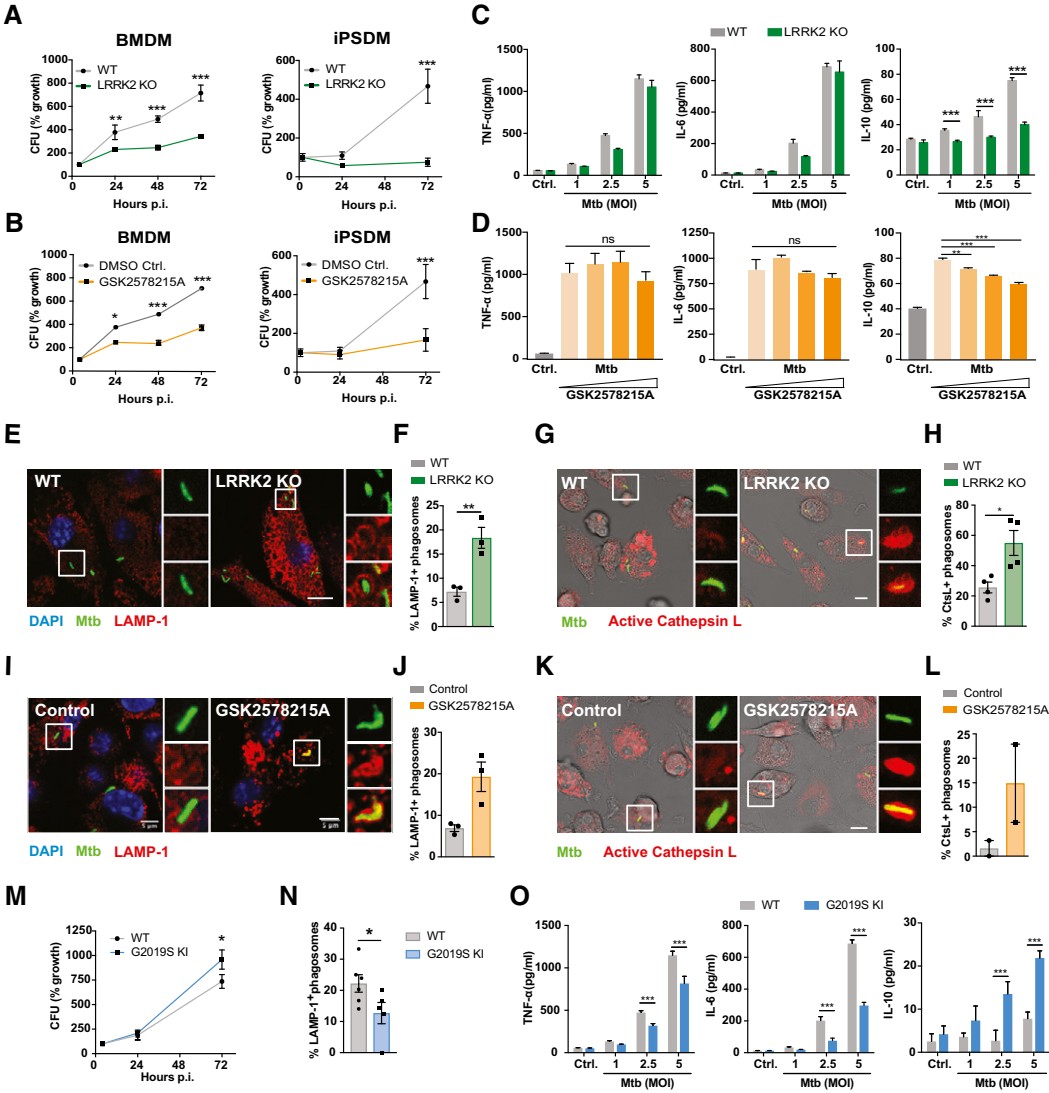

**Figure 1.  Loss of LRRK2 activity targets Mtb to phagolysosomes and limits Mtb replication.**

A    CFU from WT and LRRK2 KO mouse bone marrow-derived macrophages (BMDMs) and WT and LRRK2 KO human-induced pluripotent stem cell-derived macrophages (iPSDM). One representative experiment out of four, data show mean ± SEM of technical replicates.

B    CFU from WT BMDM and WT iPSDM treated with 1 μM GSK2578215A. For clarity, controls are the same as for panel (A). One representative experiment out of two, data show mean ± SEM of technical replicates.

C    Cytokine secretion levels in BMDM measured by ELISA at the indicated MOI. One representative experiment out of three shown.

D    Cytokine secretion levels in BMDM infected with Mtb (MOI = 5) and treated with 0.5; 1 and 3 μM GSK2578215A as measured by ELISA. One representative experiment out of two shown.

E    Representative images of WT and LRRK2 KO BMDM infected with Mtb-eGFP at 24 h after infection and stained for LAMP-1. Nuclei were labelled with DAPI. Scale bar = 10 μm.

F    Quantification of LAMP-1 co-localisation with Mtb as in panel (E). Data show three independent experiments.

G    Representative images of WT and LRRK2 KO BMDM infected with Mtb-eGFP at 24 h after infection and incubated with a substrate specific for cathepsin L as described in methods. Nuclei were labelled with DAPI. Scale bar = 10 μm.

H    Quantification of cathepsin L co-localisation with Mtb from panel (G). Data show four independent experiments.

I    Representative images of WT BMDM infected with Mtb-eGFP, treated with either DMSO (control) or 1 μM GSK2578215A at 24 h after infection and stained for LAMP-1. Nuclei were labelled with DAPI. Scale bar = 5 μm.

J    Quantification of LAMP-1 co-localisation with Mtb as in panel (I). Data show three independent experiments.

K    Representative images of cathepsin L-stained WT BMDM as in panel (I). Scale bar = 10 μm.

L    Quantification of cathepsin L co-localisation with Mtb from panel (K). Data show two independent experiments.

M    CFU from WT and G2019S LRRK2 KI BMDM. One representative experiment out of two shown.

N    LAMP-1 co-localisation with Mtb was quantified as in panels (F and J).

O    Cytokine secretion levels in BMDM measured by ELISA at the indicated MOI. One representative experiment out of two shown.

Data information: All data show mean ± SEM. *$P < 0.05$, **$P < 0.01$, ***$P < 0.001$, ns not significant. Panels (A, B, M) *t*-test adjusted for multiple comparison; panels (C, D, O) one-way ANOVA with Holm-Sidak post-test, panels (F, H, J, L, N) *t*-test.

## Loss of LRRK2 activity enhances phagosome maturation in macrophages

Because the loss of LRRK2 activity targeted Mtb to phagolyso-somes independently of autophagy, we next investigated the functional role of LRRK2 in phagosome maturation. For this, we measured the intra-phagosomal proteolysis by fluorescence-based assays with latex beads in real time (Yates & Russell, 2008) in both WT and LRRK2 KO BMDM. Notably, LRRK2 KO macrophages showed an enhanced phagosome proteolysis compared to WT macrophages (Fig 3A) without affecting phagocytic uptake (Fig EV4). In order to identify the LRRK2-dependent cellular pathways that enhanced phagosome maturation, we performed a proteomics analysis of isolated latex bead phagosomes (Trost et al, 2009) from WT and LRRK2 KO macrophages (Fig 3B, Table EV1). Strikingly, the gene ontology analysis of significantly up-regulated proteins from the LRRK2 KO phagosomes revealed a strong enrichment for proteins associated with late endocytic/lysosomal compartments and hydrolytic activity (Fig 3B, Table EV2). This analysis revealed an increase in the content of lysosomal hydro-lases such as cathepsins and lysozyme-C in LRRK2 KO phagosomes when compared to WT phagosomes (Fig 3C). In agreement with the proteomic data, Western blot analysis showed that phagosomes recruited LRRK2 and LRRK2 KO phagosomes were associated with high levels of active cathepsin D (Fig 3D). We next analysed whether LRRK2 kinase activity was required for phagosomal function by using four structurally diverse, highly specific LRRK2 kinase inhibitors, namely HG10-102-01 (Choi et al, 2012), GSK2578215A (Reith et al, 2012), LRRK2-IN-1 (Deng et al, 2011) and MLI-2 (Fell et al, 2015). All inhibitors significantly enhanced the proteolytic activity of phagosomes at different levels, indicating that the kinase activity negatively regulates phagosome maturation (Fig 3E). As expected, in macrophages from LRRK2 A2016T KI, a mutant that is active, but resistant to the inhibitors (Nichols et al, 2009), MLI-2 did not enhance phagosome proteolysis (Fig 3F). Next, we tested whether enhanced kinase activity affects phagosome function by using macrophages harbouring the PD pathogenic mutation G2019S, which enhances LRRK2 kinase activity about fourfold (West et al, 2005). Indeed, phagosomes from LRRK2 G2019S knock-in (KI) macrophages displayed reduced proteolytic activity when compared to WT macrophages (Fig 3G). This reduction in proteolytic activity observed in phagosomes from LRRK2 G2019S KI macrophages was reverted by the LRRK2 kinase inhibitor GSK2578215A in a dose-dependent manner (Fig 3H). Taken together, our data show that LRRK2 kinase activity acts as a negative regulator of phagosomal function in macrophages.

## LRRK2 activity mediates the recruitment of Class III PI3K on phagosomes

To further investigate the mechanism by which LRRK2 kinase activity enhanced phagosome maturation, we performed a label-free proteomics analysis of purified phagosomes from mouse macrophages treated or not with one of the most selective LRRK2 inhibitors, HG-10-102-01. Strikingly, we observed a remarkable change in the phagosomal proteome in response to LRRK2 kinase inhibition (Fig 4A, Table EV3). Interestingly, many proteins previously associated with LRRK2 biology were detected such as components of the

retromer complex and Rab GTPases (MacLeod et al, 2013; Steger et al, 2016). Specifically, the STRING-based protein interaction network revealed a significant reduction of the Class III phospho-inositide-3 kinase (PI3K) complex and Rubicon (RUN domain protein as Beclin-1 interacting and cysteine-rich containing), a negative regulator of the PI3K complex (Sun et al, 2010, 2011; Fig 4A). In agreement with these data, the phagosome proteome of HG-10-102-01-treated macrophages identified a highly significant decrease in total phosphatidylinositol-3-phosphate (PI3P)-binding proteins upon LRRK2 kinase inhibition (Fig 4B). We validated this result by phagoFACS (Fig EV4) using the probe eGFP-2XFYVE$_{HRS}$ that binds to PI3P and observed that the levels of eGFP-2XFYVE$_{HRS}$ on phagosomes were significantly reduced after treatment with both GSK2578215A and HG-10-102-01 in a dose-dependent manner (Fig 4C). Given that PI3K is critical for phagosome maturation (Vieira et al, 2001), we next investigated whether the low levels of this lipid kinase were linked to the LRRK2-dependent alterations in phagosome function. Notably, we could reduce PI3P levels on wild-type phagosomes using the PI3P-kinase inhibitor Vps34-IN1 (Bago et al, 2014), but not further on phagosomes from LRRK2 KO macrophages, suggesting that LRRK2 is necessary for the recruitment of this complex (Fig 4D). Consistent with a LRRK2-dependent PI3K-mediated pathway of phagosome maturation, the PI3K inhibitor LY294002 decreased proteolytic activity of phagosomes only in WT but not in LRRK2 KO macrophages (Fig 4E). Thus, LRRK2 mediates the recruitment of the PI3K complex and consequently PI3P generation on phagosomes.

## LRRK2 regulates phagosome maturation via Rubicon

We confirmed that phagosomes from LRRK2 KO macrophages showed a significant reduction of Rubicon, Beclin-1, UVRAG and Vps34 (Fig 5A). Consistent with this, inhibition of LRRK2 kinase activity reduced the recruitment of Beclin-1, Vps34 and UVRAG into phagosomes (Fig 5B and C). Whereas Vps15 levels remained unchanged, HG-10-102-01 treatment or LRRK2 KO significantly reduced Rubicon levels on phagosomes (Fig 5D). Crucially, phagosomes from LRRK2 G2019S KI macrophages showed higher levels of Rubicon when compared with WT macrophages and this effect was reverted by HG-10-102-01 (Fig 5E). Single phagosome live cell imaging analysis confirmed that Rubicon-GFP is recruited on phagosomes after PI3P association with the phagosomal membrane (Fig EV5). The proteolytic activity in phagosomes was enhanced in Rubicon KO macrophages when compared to WT macrophages (Figs 5F and EV5). Treatment of Rubicon KO macrophages with HG-10-102-01 did not further affect phagosome maturation, arguing that the phagosomal proteolytic function is driven by an LRRK2-dependent recruitment of the Rubicon complex to phagosomes (Fig 5F).

To investigate the role of Rubicon during Mtb phagosome maturation, we first confirmed its presence on the phagosomal membrane. As observed with latex beads (Fig EV5), Rubicon was recruited during the first 15 min after phagocytosis (Fig 5G). In Rubicon KD mouse macrophages (Fig 5H), the percentage of LAMP-1-positive Mtb phagosomes (Fig 5I and J) and Mtb phagosome showing pan-cathepsin activity (Schnettger et al, 2017; Fig 5K and L) was significantly higher after Rubicon KD when compared to the scrambled control. As expected, LRRK2 kinase inhibition did not further enhance LAMP-1 recruitment (Fig 5I and J) or pan-cathepsin

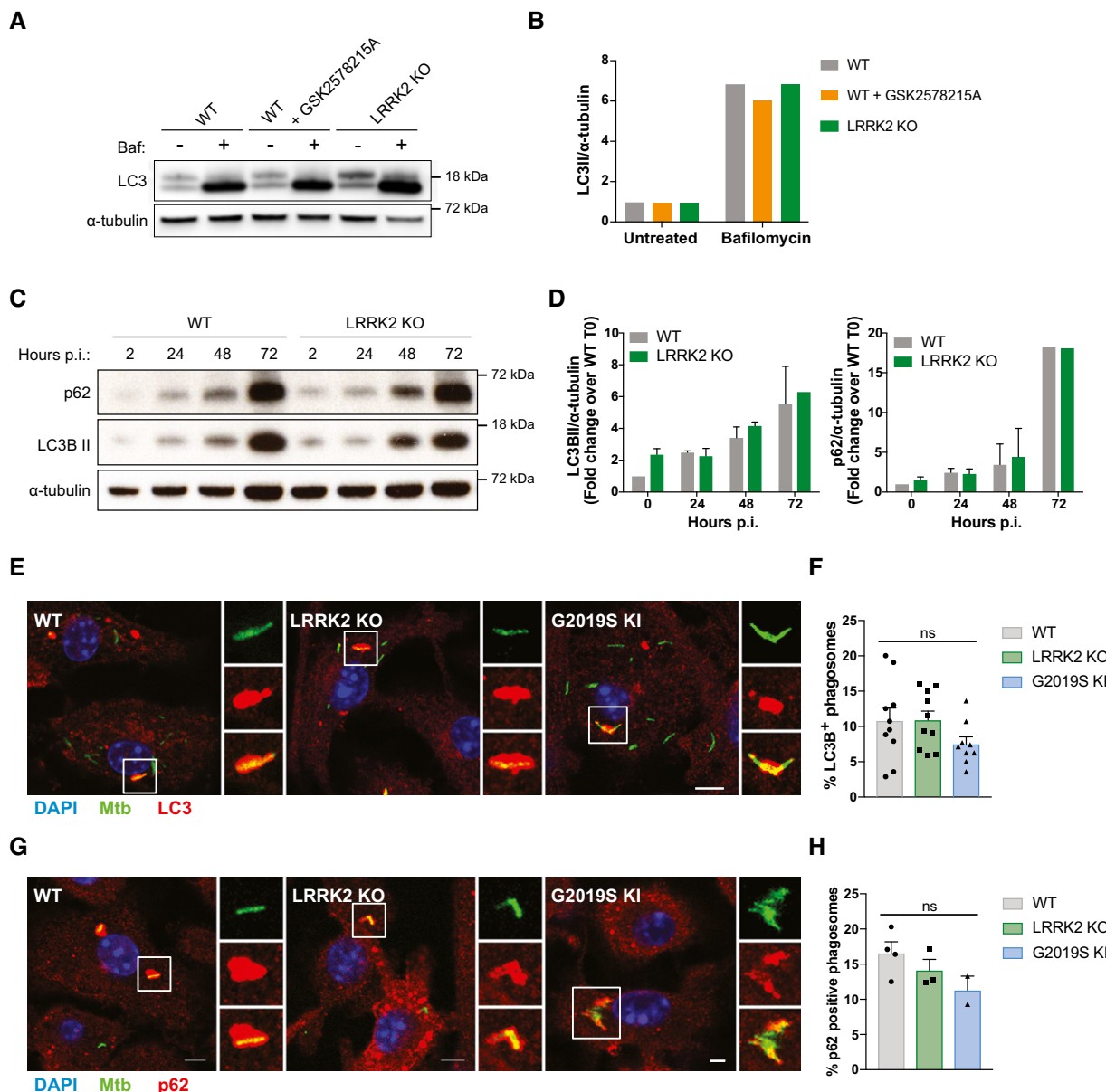

**Figure 2.   LRRK2 does not affect xenophagic targeting of *Mycobacterium tuberculosis*.**

A   WT and LRRK2 KO BMDM treated with DMSO (control) or 1 μM GSK2578215A for 2 h were treated with 100 nM Bafilomycin A1 for 4 h. Whole cell lysates were analysed by Western blotting for LC3B and α-tubulin.

B   Densitometry quantification of panel (A).

C   WT and LRRK2 KO BMDMs were infected with Mtb at MOI = 1 and whole cell lysates were analysed by Western blotting for p62, LC3BII and α-tubulin levels.

D   Quantification of panel (C). Data show mean ± SEM of three independent experiments.

E   WT, LRRK2 KO and G2019S KI BMDMs were infected with Mtb-GFP at MOI of 1 for 24 h. Recruitment of LC3B was analysed by immunofluorescence. Scale bars = 10 μm.

F   Quantitative analysis of panel (E).

G   WT, LRRK2 KO and G2019S KI BMDMs were infected with Mtb-GFP at MOI of 1 for 24 h. Recruitment of p62 was analysed by immunofluorescence.

H   Quantitative analysis of panel (G).

Data information: Data show mean ± SEM. Each dot represents an independent experiment. Data in panels (F and H) were analysed using a one-way ANOVA. ns: not significant.

activity (Fig 5K and L) of Mtb phagosomes from Rubicon-depleted macrophages. Altogether, these results show that LRRK2 kinase activity controls the recruitment of the negative regulator of PI3K, Rubicon, to phagosomes and this pathway is important for the LRRK2-dependent negative regulation of Mtb phagosome maturation.

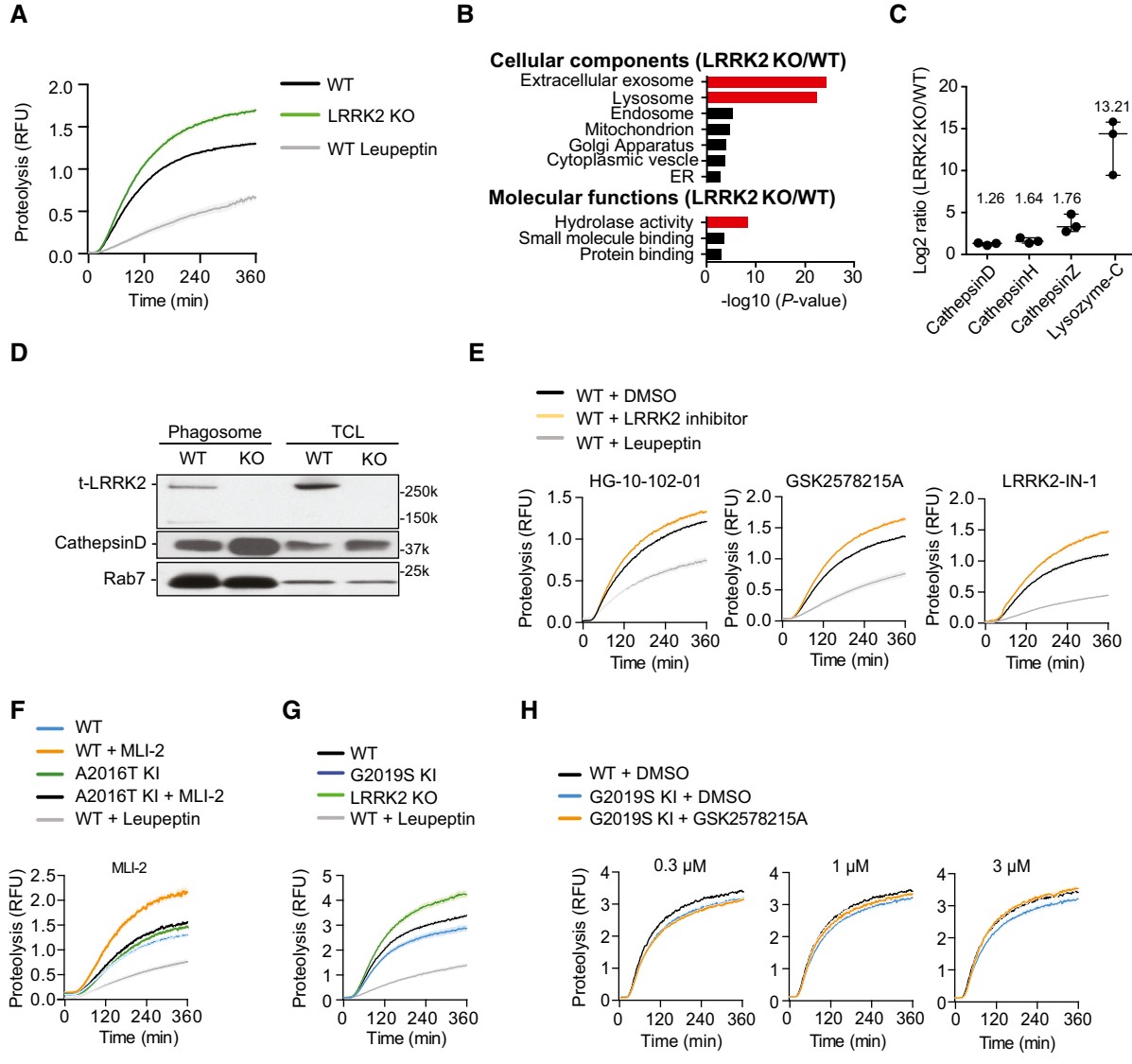

**Figure 3. Loss of LRRK2 activity enhances phagosome maturation in macrophages.**

A  Intra-phagosomal proteolysis in WT and LRRK2 KO BMDM. Cells pre-treated with 100 nM leupeptin for 1 h were used as a negative control of proteolysis. One representative experiment out of three shown.

B  Gene Ontology (GO) enrichment of cellular components and molecular functions of significantly up-regulated proteins in the proteome of LRRK2 KO-derived phagosomes compared to WT.

C  Mass spectrometry analysis of LRRK2 KO phagosomes compared to WT. Data show mean ± SEM of three biological replicates.

D  Isolated phagosomes and total cell lysates (TLC) from WT and LRRK2 KO BMDM were blotted for LRRK2, cathepsin D and Rab7 as a loading control. Data are representative of two independent experiments.

E  Intra-phagosomal proteolysis of WT BMDM pre-treated or not with 1 μM of the LRRK2 protein kinase inhibitors HG10-102-01, GSK2578215A or LRRK2-IN1. One representative experiment out of three shown.

F  Intra-phagosomal proteolysis of WT or LRRK2 A2016T KI BMDM pre-treated or not (DMSO control) with 1 μM of the LRRK2 protein kinase inhibitor MLI-2. One representative experiment out of three shown.

G  Intra-phagosomal proteolysis of WT, LRRK2 KO and LRRK2 G2019S KI BMDM. One representative experiment out of three shown.

H  Intra-phagosomal proteolysis of WT and LRRK2 G2019S KI BMDM pre-treated or not with 0.3, 1.0 and 3.0 μM GSK2578215A LRRK2 kinase inhibitor. Data show mean ± SEM and are representative of three independent biological replicates.

Data information: Shaded areas represent standard error of mean (SEM).

## Loss of LRRK2 enhances innate immunity to Mtb *in vivo*

To investigate further whether LRRK2-mediated phagosome maturation effect is associated with anti-mycobacterial defence *in vivo*, WT and LRRK2 KO mice were infected at low dose via aerosol and bacterial numbers were monitored over 56 days. At day 7 after infection with Mtb H37Rv, LRRK2 KO mice showed a significant reduction of CFUs in the lung (Fig 6A). This result was not only

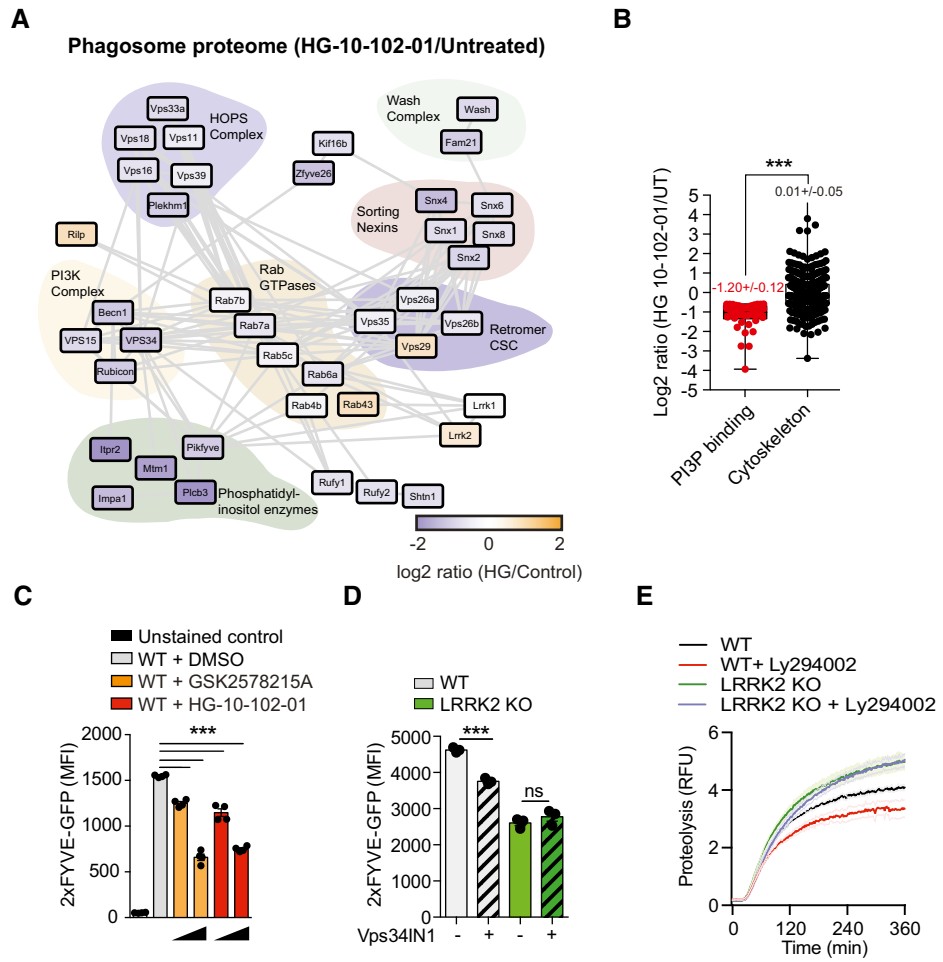

**Figure 4. LRRK2 activity mediates the recruitment of Class III PI3K on phagosomes.**

A Functional network of significantly regulated phagosome-derived proteins in response to LRRK2 protein kinase inhibition with the HG-10-102-01 inhibitor. Proteins are divided into clusters according to their molecular functions. The colour of individual proteins reflects log2 ratio in response to HG-10-102-01 treatment.

B Level of phosphatidylinositol binding proteins on phagosomes after HG-10-102-01 treatment. Cytoskeleton proteins levels were used as a control.

C PhagoFACS analysis showing levels of the PI3P-reporter eGFP-2xFYVE on RAW264.7 phagosomes pre-treated with 1 μM or 2 μM GSK2578215A or HG-10-102-01 LRRK2 inhibitors for 2 h prior bead internalisation for 30 min.

D PhagoFACS analysis showing levels of the PI3P-reporter eGFP-2xFYVE domains on RAW264.7 phagosomes pre-treated with Vps34 inhibitor Vps34IN1 in WT and LRRK2 KO phagosomes.

E Intra-phagosomal proteolysis of WT and LRRK2 KO BMDM pre-treated or not with Vps34 inhibitor LY294002.

Data information: (B–E) One representative experiment of three biological replicates shown. Data show mean ± SEM, ***$P < 0.001$, ns—not significant by Student's *t*-test.

limited to the laboratory-adapted strain H37Rv since infection with a highly virulent Beijing strain of *M. tuberculosis* N145 also showed enhanced innate immune responses in the LRRK2 KO mice (Fig 6A). Dissemination to the spleen at day 14 after infection with Mtb H37Rv was also significantly reduced in LRRK2 KO mice when compared to WT mice (Fig 6B). Surprisingly, at 56 days after infection, no significant differences in bacterial numbers in the lung and spleen were observed in mice infected with either Mtb H37Rv or N145 (Fig 6A and B). Histological analysis showed that LRRK2 KO mice had reduced number of lesions per lung and the lesions were significantly larger in LRRK2 KO than in WT mice (Fig 6C and D). Interestingly, the inflammatory cytokines IL-6, TNF-α and IFN-γ were significantly elevated in LRRK2 KO-infected mice at 56 days. We observed almost absent levels of the type I IFN-α in LRRK2 KO

lungs when compared to WT lungs from Mtb-infected mice at day 56 of infection (Fig 6E). These data suggest that the enhanced phagosome maturation observed *in vitro* results in Mtb replication restriction during the early innate immune response to Mtb in LRRK2 KO mice. These differences profoundly affected the regulation of adaptive immune responses, potentially resulting in increased T-cell activation and detrimental immunopathology.

# Discussion

Defining the mechanisms by which LRRK2 regulates cellular pathways in immune cells is critical to understand LRRK2 function. Altogether, the data presented here identified LRRK as a fine-tuner of

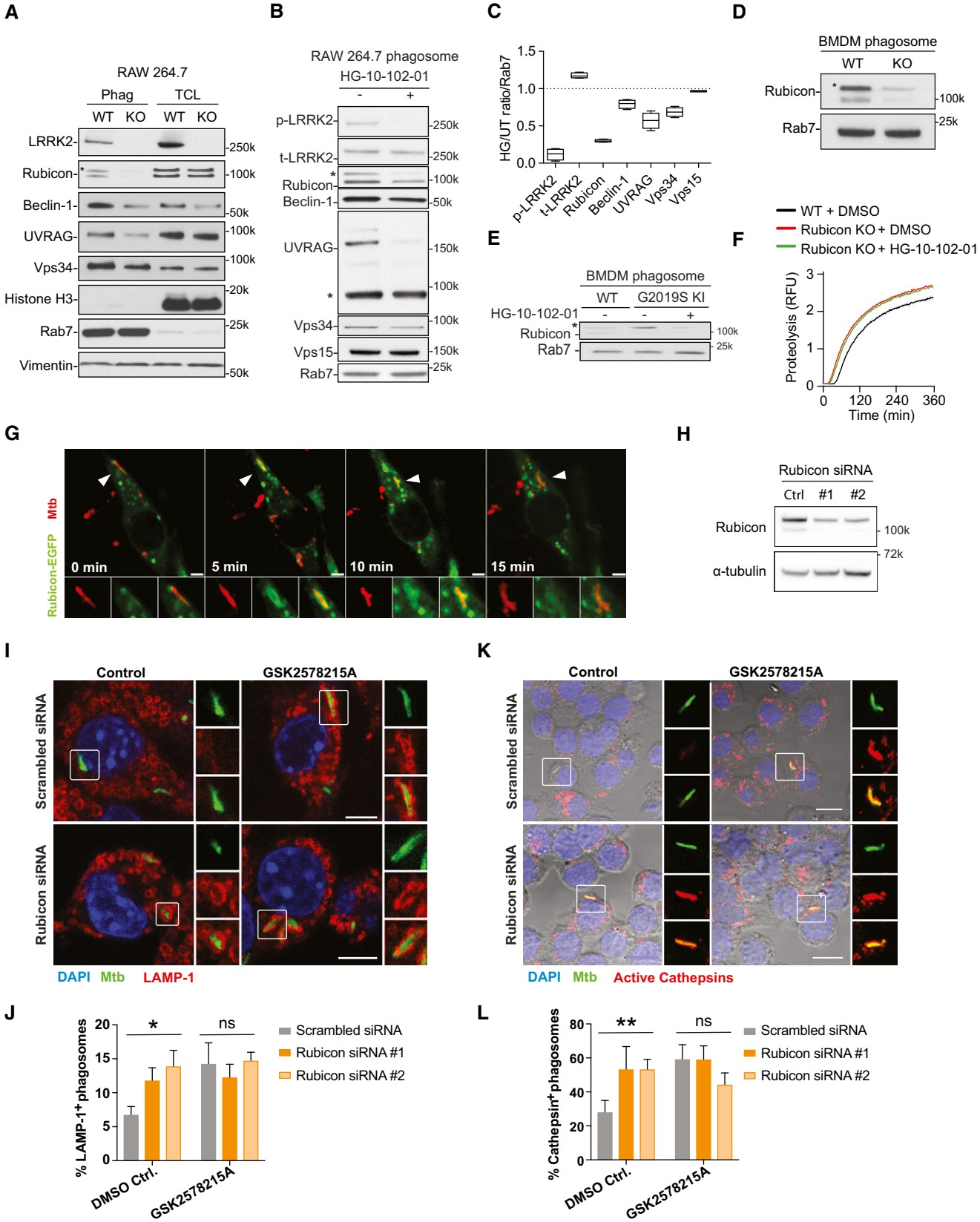

Figure 5.

**Figure 5.   LRRK2 blocks Mtb phagosome maturation via Rubicon.**

A    Western blot analysis of isolated phagosomes and total cell lysates (TCL) from WT and LRRK2 KO RAW264.7 macrophages showing LRRK2, Rubicon (130k*specific band, 100k unspecific band; see Fig EV5), Beclin-1, UVRAG and Vps34. Histone H3, Rab7 and vimentin were used as loading controls and to demonstrate purity.

B    Representative image of Western blot analysis of phagosomes isolated from RAW264.7 macrophages pre-treated or not with 2 μM HG 10-102-01 LRRK2 kinase inhibitor for 2 h showing Rubicon (130k*specific band, 100k unspecific band), Beclin-1, UVRAG and Vps34. Rab7 was used as a loading control and p-LRRK2 (Ser935) as a positive control for LRRK2 protein kinase activity.

C    Densitometry analysis of panel (B). Box and whisker plots show mean, 95% confidence interval, and maximum and minimum values of three independent experiments.

D    Western blot analysis of Rubicon levels in WT and LRRK2 KO phagosomes (* indicates specific Rubicon band; see Fig EV5).

E    Western blot analysis of Rubicon levels in WT and LRRK2 G2019S KI phagosomes pre-treated or not with 3 μM HG-10-102-01 (* indicates specific Rubicon band; see Fig EV5).

F    Intra-phagosomal proteolysis in WT and Rubicon KO BMDM pre-treated or not with 1 μM HG-10-102-01 for 1 h.

G–L   RAW264.7 macrophages were treated with siRNAs against Rubicon or non-targeting control (scrambled) and incubated or not with 1 μM GSK2578215A, followed by infection with Mtb-GFP (MOI = 1). (G) Rubicon-GFP recruitment to Mtb-RFP was monitored by live cell imaging. The arrowhead indicates the bacterium that is shown in the zoomed images. Scale bar = 5 μm. (H) Western blot analysis confirming Rubicon knock-down at 24 h post-infection. (I) Representative images of RAW264.7 macrophages pre-treated with 1 μM GSK2578215A or DMSO (control) and infected as above and stained for LAMP-1 at 24 h post-infection. Scale bar = 5 μm. (J) Quantification of LAMP-1 co-localisation from panel (I). Data show mean + SEM from one representative experiment out of three. (K) Representative images of RAW264.7 macrophages treated and infected as above and labelled with a probe for active cathepsins at 24 h post-infection. Scale bar = 10 μm. (L) Quantification of cathepsin activity (measured by BMV109) co-localisation from panel (K). Data show mean + SEM one representative experiment out of three.

Data information: (B–F) One representative experiment out of three biological replicates shown. Data show mean ± SEM. *$P < 0.05$, **$P < 0.01$ by Student's $t$-test corrected for multiple comparison. ns: not significant.

phagosome maturation by controlling PI3K activity on this organelle. Phagosome maturation is a key cellular pathway of the innate immune response, which is critical for both defending against invading pathogens and maintaining tissue homeostasis. Therefore, our data provide a mechanistic link between inflammatory diseases such as inflammatory bowel disease (Liu & Lenardo, 2012; Rocha *et al*, 2015), infectious diseases like leprosy (Zhang *et al*, 2009; Wang *et al*, 2015; Fava *et al*, 2016) and tuberculosis (Wang *et al*, 2018) to LRRK2 function.

Previous work has linked LRRK2 with phagocytosis reporting either no effect (Schapansky *et al*, 2014) or inhibition of particle internalisation (Marker *et al*, 2012). Using a different range of assays, we discovered that LRRK2 negatively regulates phagosome maturation rather than controlling internalisation. Why there are negative regulators of phagosome maturation remains unclear. However, our findings suggest that kinases such as LRRK2 are important to regulate inflammatory responses whilst efficiently targeting intracellular pathogens for degradation. Our results are consistent with the notion that LRRK2 regulates degradative pathways (preprint: Manzoni *et al*, 2015).

The improved control of Mtb replication by LRRK2 KO macrophages was unexpected since LRRK2 deficiency has been previously associated with less efficient control of *Salmonella* replication in mouse macrophages (Gardet *et al*, 2010; Zhang *et al*, 2015). However, our results with Mtb are consistent with clinical data linking specifically mycobacterial diseases and LRRK2 (Zhang *et al*, 2009; Wang *et al*, 2015, 2018; Fava *et al*, 2016). Considering that both PI3K and PI3P are critical for phagosome maturation (Vieira *et al*, 2001), our data argue that LRRK2 regulates phagosome maturation by modulating PI3P levels on phagosomes. Because PI3K activity is linked to various cellular processes, our data suggest that LRRK2, by controlling the localisation of PI3K on phagosomes, regulates the function of this organelle. Importantly, our studies established a novel mechanistic link between LRRK2 and Rubicon/PI3K complex that may explain LRRK2 functions in the context of endocytosis, exocytosis and autophagocytosis (preprint: Manzoni *et al*, 2015).

A previous study linked PD and TB at the cellular level showing that the ubiquitin ligase PARKIN has been implicated in both PD and TB control (Manzanillo *et al*, 2013). In contrast to PARKIN, the loss of LRRK2 resulted in improved Mtb control by macrophages highlighting that the cross talk between common pathways regulating immunity and PD susceptibility is multifaceted. Collectively, this work uncovers a ubiquitin-independent PI3K-dependent pathway that implicates the most common gene associated with genetic PD, LRRK2, in one of the most important functions of macrophages in tissue homeostasis and innate immunity. Because the process of phagosome maturation is also important for inflammation, our results also suggest LRRK2 as a regulator of the balance between pro- and anti-inflammatory responses in both TB and PD.

Importantly, we found that the loss of LRRK2 enhances early innate immune responses to Mtb *in vivo* with potential consequences in the cross talk of innate and adaptive immunity later during the infection. Although more studies are required to better understand this LRRK2-dependent cross talk in PD, the elevated levels of TNF-α observed at later stages of infection are in agreement with reports showing high concentrations of TNF-α in the brain and Cerebrospinal fluid (CSF) of PD patients (Mogi *et al*, 1994) and deregulation of inflammation in the context of bacterial infections (Liu *et al*, 2011; Fava *et al*, 2016). After infection with Mtb, we observed that in LRRK2 KO mice, the transcriptional levels of type II IFN (IFN-γ) were significantly higher and levels of type I IFN (IFN-α) almost absent. We hypothesise that LRRK2 could be one of the molecules predicted to act at very early stages of the infection (Verrall *et al*, 2014). However, later during the infection, there is a recruitment of T cells in the lungs that can compensate for the outcome of the infection. This could explain why in the LRRK2 KO-infected mice bacterial burdens are similar in the presence of a pro-inflammatory cytokine profile. Moreover, it is becoming evident that LRRK2 could have a function in T cells, so our study opens this possibility for future investigations. Interestingly, infection of IFNAR KO mice with Mtb shows a similar phenotype to the LRRK2 KO mice. In IFNAR KO mice, Mtb replication restriction was better than in WT mice at early time points, but no differences in bacterial

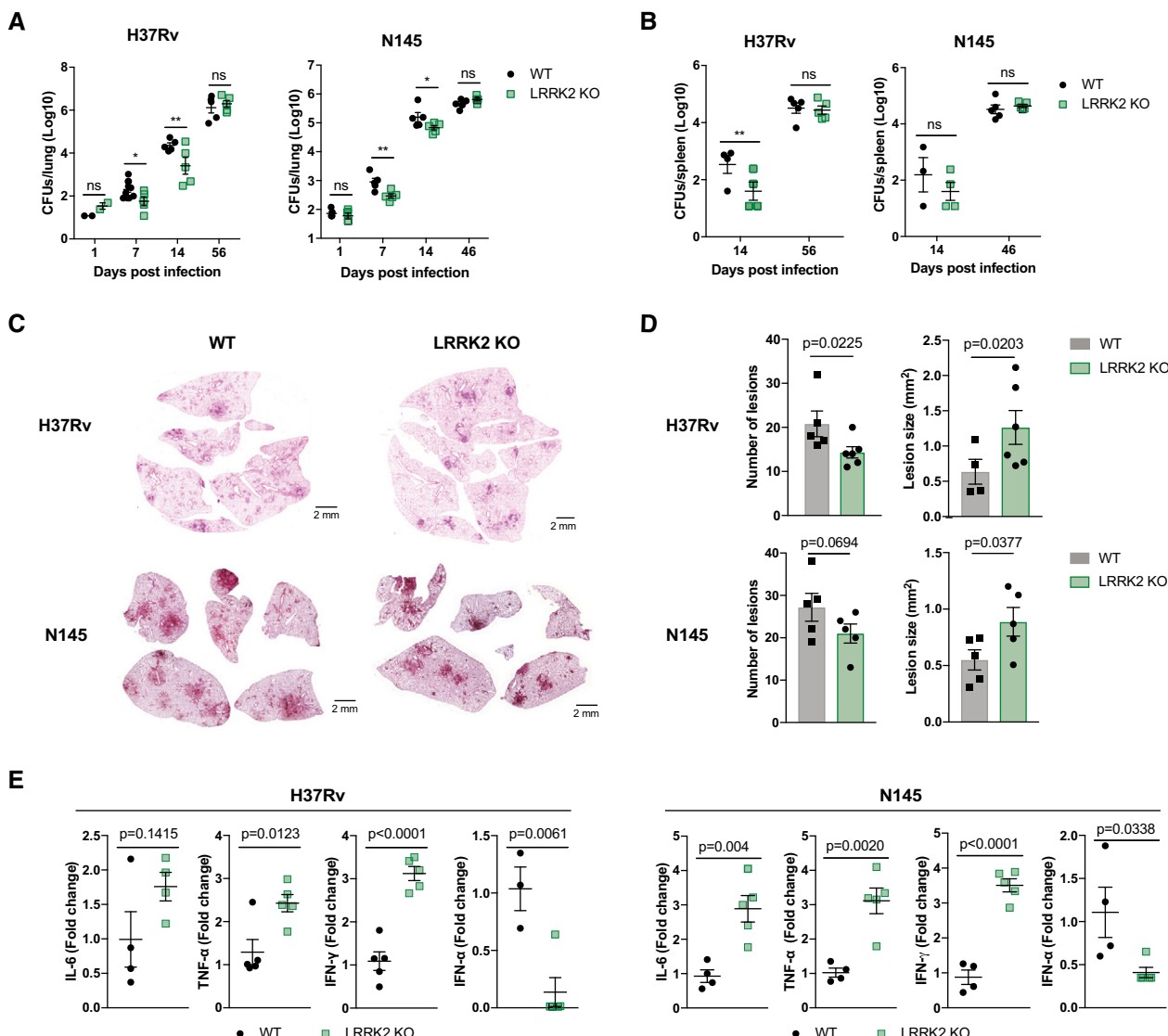

**Figure 6.  Loss of LRRK2 enhances innate immunity to Mtb *in vivo*.**

A   WT and LRRK2 KO mice were infected with a low dose of *Mycobacterium tuberculosis* strain H37Rv or N145. Growth of *M. tuberculosis* H37Rv or N145 in lungs was measured by CFU.

B   Same as in panel (A) in spleens.

C   Haematoxylin- and Eosin-stained lung sections of one representative animal per genotype and Mtb strain on day 56 after infection.

D   Quantification of the number of lesions and lesion area in infected lungs (*n* = 5).

E   mRNA levels of cytokines were determined from whole lung homogenates 56 days post-infection (*n* = 5).

Data information: All data represent mean ± SEM. Groups were compared using an unpaired *t*-test, and all *P*-values are corrected for multiple comparisons using the Holm-Sidak method. *P < 0.05, **P < 0.01, ns = not significant. Each dot represents an individual animal.

numbers were observed at later time points (Kimmey *et al*, 2017). Therefore, it seems that the IFN/IL-10 signalling axis in general has a bi-phasic effect on innate and adaptive immune responses to Mtb.

The observed differences in cytokine profiles in LRRK2 KO mice after Mtb infection highlight that LRRK2 controls specific inflammatory pathways which need to be considered when evaluating the long-term use of LRRK2 kinase inhibitors in PD patients. Phagocytosis and degradation of dying neurons by microglial cells are recognised as critical for the pathogenesis of PD. The LRRK2-dependent

deregulation of phagosome function could therefore be important during microglial cell activation after neuronal apoptosis and subsequently contribute to neurodegeneration and neuroinflammation. Taken together, our study identified that LRRK2 negatively regulates phagosome maturation by recruiting Rubicon/PI3K complex to phagosomes thereby altering downstream cytokine signalling. Importantly, these data uncover LRRK2 as a regulator of the early clearance of Mtb, suggesting the kinase activity of LRRK2 can be a potential target for host-directed therapies in tuberculosis.

# Materials and Methods

### Reagents

GSK2578215A (Reith *et al*, 2012) was obtained from Tocris or GlaxoSmithKline; MLI-2 (Fell *et al*, 2015) was obtained from Merck; Bafilomycin A1 (#B1793) was obtained from Sigma, and HG-10-102-01 (Choi *et al*, 2012), LRRK2-IN1 (Deng *et al*, 2011) and Vps34-IN1 (Bago *et al*, 2014) were custom synthesised by Natalia Shapiro (University of Dundee, UK).

### Antibodies

Antibodies against Rubicon (clone D9F7, #8465), Vps34 (clone D9A5, #4263), Vps15 (#14580), p62 for Western blotting (#5114), α-tubulin (clone DM1A, #3873) and Rab7 (clone D95F2, #9367) were from Cell Signaling, anti-cathepsin-D (#ab75852) and anti-LC3B for Western blotting (#ab48394) from Abcam, anti-LAMP1 (clone 1D4B) from DSHB, anti-p62 (#111393) for IF from GeneTex and anti-LC3B (PM036) for IF from MBL. Rabbit monoclonal antibodies for total LRRK2 (t-LRRK2) and pS935-LRRK2 (p-LRRK2; Dzamko *et al*, 2012), sheep anti-Rubicon, anti-UVRAG and anti-Beclin-1 were generated at the University of Dundee. Recombinant proteins, plasmids and antibodies generated at the University of Dundee for the present study are available to request on our reagents website (https://mrcppureagents.dundee.ac.uk/).

### Mice

C57BL/6J (WT) mice were purchased from the Jackson Laboratory. LRRK2 KO (B6.129 × 1(FVB)-LRRK2$^{tm1.1Cai}$/J) were either purchased from the Jackson Laboratory or were a kind gift of Huaibin Cai (NIH, US). LRRK2 G2019S knock-in mice (Steger *et al*, 2016) were provided by GSK (GSK Stevenage, UK). LRRK2 A2017T (Steger *et al*, 2016) mice were from Dario Alessi (University of Dundee, UK). Mice were maintained under specific pathogen-free conditions at the University of Dundee (UK) or the Francis Crick Institute (UK). Animal studies and breeding were approved by the University of Dundee or Francis Crick Institute ethical committee and performed under U.K Home Office project licence (PPL 70/8045 & 60/4387). All animal studies were ethically reviewed and carried out in accordance with Animals (Scientific Procedures) Act 1986 and the GSK Policy on the Care, Welfare and Treatment of Animals.

### Mouse macrophages

Bone marrow-derived macrophages (BMDMs) were derived from femurs and tibiae of female wild-type and mutant C57BL/6 mice between 8 and 12 weeks of age. Rubicon KO mouse was described before (Martinez *et al*, 2015). Bone marrow was extracted and differentiated into macrophages in RPMI or DMEM (Life Technologies, Gibco) containing 10% heat-inactivated foetal calf serum (FCS, Life Technologies, Gibco) and 20% L929-conditioned medium for 7 days. The murine macrophage cell line RAW264.7 was obtained from ATCC and cultured in DMEM (Life Technologies, Gibco) supplemented with 10% heat-inactivated FCS. iPS-derived macrophages were generated as described before (van Wilgenburg *et al*, 2013).

### Human iPSC-derived macrophages

The human biological samples were sourced ethically, and their research use was in accord with the terms of the informed consents. The human iPSC line SFC840-03-03 (AH017-13) has been published previously (Fernandes *et al*, 2016). It was derived from a disease-free control donor in the Oxford Parkinson's Disease Centre cohort, having given signed informed consent, which included mutation screening and derivation of hiPSC lines from skin biopsies (Ethics committee: National Health Service, Health Research Authority, NRES Committee South Central—Berkshire, UK, who specifically approved this part of the study—REC 10/H0505/71). Gene editing of this line was carried out using CRISPR/Cas9-mediated disruption of exon 3 using a double nickase strategy (to avoid the possibility of off-target cleavage) with guide RNAs inserted into plasmid px462 [Addgene (Ran *et al*, 2013)]. Clones were picked and screened for bi-allelic out-of-frame repair by high resolution melt analysis, confirmed by sequencing, and SNP-karyotyped to confirm no chromosomal abnormalities. Clone SFC840-03-03 LRRK2$^{-/-}$ D10 was used for this study. Macrophages were derived from the edited and control iPSC as previously published (van Wilgenburg *et al*, 2013) and for the final stage were differentiated in X-VIVO 15 media without gentamicin (Lonza) containing GlutaMAX (Gibco) and 100 ng/ml M-CSF (Thermo Fisher) for 1 week. At day 3 of differentiation, 50% of the medium was replaced with fresh medium.

### *Mycobacterium tuberculosis* strains and culture

*Mycobacterium tuberculosis* H37Rv, H37Rv-eGFP or RFP and N145 were cultured in Middlebrook 7H9 broth (M0178, Sigma-Aldrich) supplemented with 10% Middlebrook ADC (212352, BD), 0.05% Tween-80 and 0.004% glycerol. Bacteria were incubated at 37°C with constant rotation. All bacteria were used at mid-exponential phase (OD$_{600}$ from 0.6 to 0.8) for infection experiments.

### Phagosomal proteolysis assays

Fluorogenic assays for phagosomal proteolysis were adapted from the method from the Russell laboratory (Yates *et al*, 2005; VanderVen *et al*, 2010; Podinovskaia *et al*, 2013). BMDMs were plated onto 96-well plates at $1 × 10^5$ cells/ml 24 h prior to the experiment. DQ red BSA (Life Technologies)-coupled carboxylated silica beads (3 μm, Kisker Biotech) were diluted 1:200 in binding buffer (1 mM CaCl$_2$, 2.7 mM KCl, 0.5 mM MgCl$_2$, 5 mM dextrose, 10 mM hydroxyethyl piperazine ethane sulphonate (HEPES) and 5% FCS in PBS pH 7.2) and incubated with BMDM for 3 min at room temperature. Beads were replaced with warm binding buffer, and real-time fluorescence was measured at 37°C using a SpectraMax Gemini EM Fluorescence Microplate Reader (Molecular Devices), set as maximal readings per well to allow reading time intervals of 2 min. Plots were generated from the ratios of signal/control fluorescence.

### Phagosome isolation

Phagosomes were isolated as described previously (Desjardins *et al*, 1994; Hartlova *et al*, 2017; Peltier *et al*, 2017; Trost *et al*, 2009). Briefly, 1.0 μm carboxylated polystyrene beads (Merck-Millipore)

were incubated with macrophage monolayers for 30-min pulse. Subsequently, cells were washed and lysed, nuclei and cell debris removed, and phagosomes isolated using a sucrose gradient.

## Proteomic sample preparation and dimethyl labelling

Proteins from isolated phagosomes were solubilised in 1% sodium 3-[(2-methyl-2-undecyl-1,3-dioxolan-4-yl)methoxy]-1-propanesulfonate(commercially available as RapiGest, Waters) in 50 mM Tris–HCl pH 8.0 with 5 mM Tris (2-carboxyethyl) phosphine (TCEP, Pierce), heated at 60°C for 5 min. Proteins were then alkylated in 10 mM iodoacetamide (Sigma), and excess reagent was quenched with 10 mM dithiothreitol (DTT, Sigma). As extracted protein amount varied among the various ligand types, 4 μg of protein was processed for each sample, as quantified by EZQ (Life Technologies). After 10-fold dilution in 50 mM Tris pH 8.0, 5 mM calcium chloride, Trypsin Gold (Promega) was added at 1:100 for 4 h at 37°C with shaking, and an additional dose for overnight incubation. Sodium 3-[(2-methyl-2-undecyl-1,3-dioxolan-4-yl)methoxy]-1-propanesulfonate was removed by adding 1% trifluoroacetic acid heated at 37°C for 1 h and centrifugation at $14,000 \times g$ for 30 min. Peptides were desalted by solid phase extraction using Macrospin C-18 (Harvard Apparatus), lyophilised and either labelled using isotopically labelled formaldehyde (Boersema *et al*, 2008) or left unlabelled for label-free experiments. Peptides were checked for labelling efficiency of lysines and N-termini (> 98%).

## LC-MS/MS and protein identification

Label-free mass spectrometric samples were analysed on an Ultimate 3000 Rapid Separation LC Systems chromatography (Thermo Scientific) with a C18 PepMap, serving as a trapping column (2 cm × 100 μm ID, PepMap C18, 5-μm particles, 100 Å pore size) followed by a 50 cm EASY-Spray column (50 cm × 75 μm ID, PepMap C18, 2-μm particles, 100 Å pore size; Thermo Scientific) with a linear gradient of 2.4–35% (ACN, 0.1% FA) over 325 and 30 min (Phagosomes) or 180-min (IP) washing and re-equilibration step at 300 nl/min. Mass spectrometric identification was performed on an Orbitrap Fusion Tribrid mass spectrometer (Thermo Scientific) operated in "Top Speed" data-dependant mode, operated in positive ion mode. FullScan spectra were acquired in a range from 400 m/z to 1,600 m/z, at a resolution of 120,000 (at 200 m/z), with an automated gain control (AGC) of 300,000 and a maximum injection time of 50 ms. Charge state screening is enabled to exclude precursors with a charge state of 1. The intensity threshold for a MS/MS fragmentation is set to $10^4$ counts. The most intense precursor ions are isolated with a quadrupole mass filter width of 1.6 m/z, and CID fragmentation was performed in one-step collision energy of 32% and activation Q of 0.25. MS/MS fragments ions were analysed in the segmented linear ion trap with a normal scan range, in a rapid mode. The detection of MS/MS fragments was set up as the "Universal Method", using a maximum injection time of 300 ms and a maximum AGC of 2,000 ions. Dimethyl-labelled samples were analysed as follows: mass spectrometric analyses were conducted similarly as previously described (Ritorto *et al*, 2013; Dill *et al*, 2015; Guo *et al*, 2015). In detail, biological triplicates or quadruplicates of mixes of 1 μg of light-labelled and 1 μg of heavy-labelled samples were analysed on an Orbitrap Velos Pro mass spectrometer

coupled to an Ultimate 3000 UHPLC system with a 50 cm Acclaim PepMap 100 or Easy-Spray analytical column (75 μm ID, 3 μm C18) in conjunction with a Pepmap trapping column (100 μm × 2 cm, 5 μm C18; Thermo Fisher Scientific). Six hour linear gradients were performed from 3% solvent B to 35% solvent B (solvent A: 0.1% formic acid, solvent B: 80% acetonitrile 0.08% formic acid) with a 30-min washing and re-equilibration step. Mass spectrometer acquisition settings were as follows: lockmass of 445.120024, MS1 with 60,000 resolution, top 20 CID MS/MS using Rapid Scan, mono-isotopic precursor selection, unassigned charge states and $z = 1$ rejected, dynamic exclusion of 60s with repeat count 1.

## Protein identification and quantification

Protein identification and label-free quantification were performed using MaxQuant Version 1.5.1.7 (Cox & Mann, 2008) with the following parameters: stable modification carbamidomethyl (C), variable modifications oxidation (M), acetylation (protein N-terminus) and deamidation (NQ). Search was conducted using the Uniprot-Trembl *Mus musculus* database (42,095 entries, downloaded March 17th, 2015) including common contaminants. Mass accuracy was set to 4.5 ppm for precursor ions and 0.5 Da for ion trap MS/MS data. Identifications were filtered at a 1% false-discovery rate (FDR) at the protein level, accepting a minimum peptide length of five amino acids. Label-free quantification of identified proteins referred to razor and unique peptides and required a minimum ratio count of 2. Normalised ratios were extracted for each protein/condition and were used for downstream analyses. A Student's *t*-test (two-tailed, homoscedastic) was performed on the calculated ratios, and proteins with $P < 0.05$ and a fold change > twofold were considered significantly altered in abundance.

## DAVID annotation clustering network

Significant mass spectrometry hits (> 1.5-fold change, $P < 0.05$) were classified by DAVID (Database for Annotation, Visualization, and Integrated Discovery) Bioinformatics Resources (v6.8; Huang *et al*, 2007) where proteins are assigned in gene ontology (GO) terms, which rely on a controlled vocabulary for describing a protein in terms of its molecular function and cellular component.

## STRING network analysis

Significant mass spectrometry hits (> 1.5-fold change, $P < 0.05$) were loaded into the String database v10.0 (Szklarczyk *et al*, 2015), and the network was exported using default parameters. The networks were exported into Cytoscape (Shannon *et al*, 2003).

## Western blotting

For lysis, cells or phagosomes were washed twice on ice with PBS, directly lysed in 2× LDS sample buffer and protein concentration was determined by EZQ assay (Invitrogen). The samples were boiled at 70°C for 10 min together with sample buffer and reducing agent (NuPAGE, Life Technologies) and run on a NuPAGE 4–12% Bis-Tris gel (Life Technologies). The gels were transferred onto a

PVDF membrane using the Mini Trans-Blot Cell system from Bio-Rad. The membranes were blocked in 5% semi-skinned milk in TBS-T (TBS, 0.1% Tween-20 (MP Biomedicals LLC)). The membranes were incubated with primary in 3% BSA in TBS-T at 4°C overnight and with the secondary antibodies in 5% skimmed milk in PBS-T for 1 h at room temperature. Western blots were quantified by densitometry using ImageJ.

### PhagoFACS

For selective PI3P staining on the phagosomes, the 2×FYVE domain of HRS fused to green fluorescent protein (GFP–2×FYVE$_{Hrs}$) was utilised. For phagosome staining, RAW264.7 cells were pre-treated for 2 h with 1 μM or 2 μM GSK2578215A and/or HG-10-102-01 prior pulsing cells with 1-μm latex beads for 30 min. After homogenisation in hypotonic buffer (3 mM imidazole, 250 mM sucrose and inhibitor of proteases and phosphatases), the post-nuclear fraction containing phagosomes was fixed with 1% paraformaldehyde (PFA) in PBS for 10 min on ice. Fixation was stopped by the addition of PBS-glycine at a final concentration of 0.2 M. The post-nuclear supernatants were incubated for 30 min at RT with 5 μg/ml FYVE domain conjugate in PBS supplemented with 0.1% bovine serum albumin (BSA). Preparations were analysed after gating on a particular FSC/SSC region (corresponding to single beads in a solution). Samples were acquired on a BD FacsCanto II flow cytometer and analysed by FlowJo software v10.

### Infection of macrophages with *Mycobacterium tuberculosis*

Bacterial cultures were pelleted by spinning for 5 min at 2,000 × *g* and washed twice with RPMI containing 10% heat-inactivated FCS. Sterile 2.5- to 3.5-mm glass beads were added at a volume equal to the bacterial pellet size. The tube was vigorously shaken for 1 min to break up bacterial clumps. The bacteria were resuspended in RPMI containing 10% heat-inactivated FCS and centrifuged at 300 × *g* for 5 min. The supernatant was transferred into a fresh tube, and bacterial numbers were estimated by measuring the OD$_{600}$ assuming that OD 0.1 contains $1 \times 10^7$ bacteria. Bacteria were added at a MOI = 0.5 for BMDMs and MOI = 5 for RAW264.7 cells in the presence of inhibitors. The infection was allowed to proceed for 2 h, after which all cells were washed once with PBS and the medium was replaced by RPMI containing 10% heat-inactivated FCS.

### Indirect immunofluorescence

For immunofluorescence, $1 \times 10^5$ RAW264.7 cells or $2 \times 10^5$ BMDM were seeded on coverslips and rested overnight. If LRRK2 kinase inhibition was required, macrophages were treated with GSK2578215A inhibitor at 1 μM for 2 h before infection. At the indicated time points, cells were fixed with 4% methanol-free PFA (15710, Electron Microscopy Sciences) in PBS for 24 h at 4°C. Coverslips were then quenched with 50 mM NH$_4$Cl (A9434, Sigma-Aldrich) in PBS for 10 min at room temperature and permeabilised with 0.3% Triton X-100, 5% FCS in PBS for 15 min. The primary antibody was diluted in PBS containing 5% FCS and incubated for 1 h at RT. The coverslips were washed three times in PBS, and the secondary antibody was added in the same way as

the primary antibody (anti-rat or rabbit-Alexa fluor 488 or Alexa fluor 568, Invitrogen) for 45 min at room temperature. After three more washes with PBS, nuclear staining was performed using 300 nM DAPI (Life Technologies, D3571) in PBS for 10 min. One final wash with PBS was performed before mounting the coverslips on glass slides using DAKO mounting medium (DAKO Cytomation, S3023). Images were acquired on a Leica SP5 inverted microscope. Images were analysed using the image analysis software ImageJ. Association of LAMP-1, LC3B and p62 was quantified by creating a mask of the bacterial outline, widen the mask by two pixels, and measuring mean fluorescence intensity of the marker in the masked area.

### Cathepsin activity assay

For measuring cathepsin activity in macrophages, the cathepsin-L MAGIC RED™ substrate (ICT941, Bio-Rad) or the pan-cathepsin probe BMV109 (BioVergent) was added at a final dilution of 1:300 or 1:1,000, respectively, and incubated for 20 min at 37°C prior to imaging. The cells were washed twice with PBS and imaged directly in RPMI containing 10% heat-inactivated FCS at 37°C, 5% CO$_2$.

### Rubicon association with Mtb phagosomes

For dynamic association of Rubicon-GFP to Mtb-RFP, RAW264.7 macrophages were plated at $1 \times 10^5$ on WillCo-dish® glass-bottom dishes and transfected with 1 μg plasmid DNA using Lipofectamine 3000 (Thermo Fisher Scientific) 1 day prior to infection. Imaging was performed using a Leica TCS SP5 II microscope (Leica Microsystems) equipped with AOBS, a HC PLAOP CS2 63.0 × 1.40 OIL objective and an environmental control chamber providing 37°C, 5% CO$_2$ and 20–30% humidity.

### Rubicon knock-down

RAW264.7 cells were seeded at $1 \times 10^5$ cells per 24 well and treated with anti-Rubicon OnTargetPlus siRNA smart-pool (L-172564-00-0005, Dharmacon; siRNA#1), anti-Rubicon siRNA Silencer Select (4390771, Ambion; siRNA#2) or recommended scrambled control (Dharmacon). siRNA was diluted in 50 μl RPMI, and 3 μl HiPerfect (Qiagen) was diluted in 50 μl RPMI. The two mixtures were combined, resulting in a 200 nM siRNA transfection stock. The transfection complexes were added to the cells at a final concentration of 50 nM. After 24 h, cells were infected with *M. tuberculosis*. Knock-down was confirmed throughout the infection period by Western blot.

### CFU analysis

At the indicated time points, the cells were washed once with PBS and then lysed in 500 μl sterile water containing 0.05% Tween-80 to release intracellular bacteria. After 45 min of incubation at room temperature, the lysed cell solution was serially diluted in 10-fold steps in PBS containing 0.05% Tween-20 and plated onto 7H11 agar plates as described above. For calculation of the percentage of growth, the 2-h time point was considered 100%, normalising all following time points to this time point. For CFU analysis, bacteria were cultured on Middlebrook 7H11 (M0428, Sigma-Aldrich) agar

plates supplemented with 10% Middlebrook OADC (212351, BD) and 0.05% Tween-80 and incubated at 37°C for 3–4 weeks.

## ELISA

Cells were seeded in a 96-well plate and infected with *M. tuberculosis* at a MOI of 1–5. The supernatant was collected at 24 h post-infection and filtered twice through a 0.22-μm PVDF membrane. ELISAs for TNF-α, IL-6 and IL-10 (DuoSet mouse ELISA kits from R&D Systems) were performed according to manufacturer's instruction.

## Murine infections

C57BL/6J (WT) mice and LRRK2 KO (B6.129 × 1(FVB)-LRRK2$^{tm1.1Cai}$/J) were housed under specific pathogen-free condition at the Francis Crick Institute. All animals were bred and maintained in accordance with the United Kingdom Home Office regulations, and all experimental protocols were carried out under the project licence 70/8045. Groups of 6- to 8-week-old female mice were infected by low-dose aerosol exposure with a mid-log phase culture of Mtb H37Rv or N145 using a Glas-Col (Terre Haute, IN) aerosol generator. The aerosoliser was calibrated to deliver approximately 200 bacteria into the lungs. Bacterial counts in the lungs at each time point of the study were determined by plating serial dilutions of individual organ homogenates on duplicate plates of Middlebrook 7H11 agar containing OADC enrichment. CFUs were counted after 3- to 4-week incubation at 37°C as described in CFU analysis.

## Histology

Lungs were perfused in 4% PFA for a minimum of 24 h followed by transfer to 70% ethanol. Lungs were paraffin embedded, section and stained for haematoxylin and eosin (H&E). Lung sections were scanned with an Olympus Virtual Slide Microscope VS120 equipped with a 40× objective. Lung inflammation in Mtb-infected mice was analysed by measuring inflamed dense area in two lung sections per mouse using the image analysis software ImageJ.

## Cytokine mRNA expression in Mtb-infected mice

Whole lungs were homogenised in TRIzol® Reagent (Thermo Fisher), and RNA was extracted using chloroform. RNA was reverse transcribed using the QuantiTect Reverse Transcription Kit (QIAGEN) according to the manufacturer's protocol. Murine TNF-α, IL-6, IFN-γ and GAPDH were detected using TaqMan probes (Thermo Fisher), and real-time PCRs were run on a QuantStudio™ 6 Flex Real-Time PCR System. Fold change expression was calculated using the $\Delta\Delta C_t$ method normalised to WT.

## Image analysis

Images were analysed using the image analysis software FIJI (US National Institutes of Health, USA). Marker association with Mtb was analysed as previously described (Schnettger *et al*, 2017). In brief, association was measured by automated analysis of the mean relative fluorescence intensity in a two-pixel wide ring around the

bacteria. The mean intensity defining a marker-positive bacterium was determined empirically and used as a threshold to calculate the percentage of positive bacteria. At least 100 bacteria per biological replicate were analysed during the analysis.

## Statistical analysis

Statistical analysis was performed using GraphPad Prism software. Definition of statistical analysis and *post hoc* tests used can be found in figure legends. The statistical significance of data is denoted on graphs by asterisks (*) where $*P < 0.05$, $**P < 0.01$, $***P < 0.001$ or ns = not significant.

## Data availability

The mass spectrometry proteomics data have been deposited to the ProteomeXchange Consortium via the PRIDE (Vizcaino *et al*, 2016) partner repository with the dataset identifier PXD006909.

**Expanded View** for this article is available online.

## Acknowledgements

We thank the DNA cloning, Protein & Antibody Production, DNA sequencing facility, tissue culture and mass spectrometry teams of the MRC PPU, Xiao Zou for preliminary experiments, Natalia Shapiro for synthesis of inhibitors, Carol Clacher; Laura Frew and Gail Gilmour in Transgenic Services (University of Dundee); Isobel Stula and *In Vivo* Science and Delivery team (GSK, Stevenage) for breeding and supply of the LRRK2 G2019S KI mice to MGG Lab and Sebastien Gagneux for Mtb strains. This work was funded by Medical Research Council UK (MR/N026004/1 and MR/L010933/1 to PAL, MC_UU_12016/5 to MT and MC_UP_1202/11 to MGG); the pharmaceutical companies supporting the Division of Signal Transduction Therapy Unit (Boehringer Ingelheim, GSK and Merck KGaA, to MT); the Michael J. Fox Foundation (to MGG and PAL), Parkinson's UK Fellowship F1002 to PAL and the Francis Crick Institute (to MGG), which receives its core funding from Cancer Research UK (FC001092), the Medical Research Council UK (FC001092) and the Wellcome Trust (FC001092); the Oxford Martin School (LC0910-004) and the Wellcome Trust (WTISSF121302) to SAC; the Innovative Medicines Initiative Joint Undertaking (IMI JU, 115439), the European Union's Seventh Framework Programme, EFPIA companies' in kind contribution to SAC. This publication reflects only the author's views and neither the IMI JU nor EFPIA nor the European Commission are liable for any use that may be made of the information contained therein (SAC and RF), the Monument Trust Discovery Award from Parkinson's UK, the National Institute for Health Research (NIHR) Oxford Biomedical Research Centre and the NIHR Comprehensive Local Research Network.

## Author contributions

AH and SH performed most experiments; OB-G and AR performed experiments; JP and BDD performed proteomics experiments; AH, SH, JP, MT, AF, MGG and BDD performed data analysis; SAC, HL, RF, PD, PAL, JM, IGG, DRA and ADR provided intellectual input and tools; MT, MGG, AH and SH designed experiments; AH, SH, MT and MGG wrote the manuscript with contributions from all authors.

## Conflict of interest

ADR is an employee of GlaxoSmithKline, a global healthcare company that may conceivably benefit financially through this publication.

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
