## [Review Process File · The EMBO Journal]

LRRK2 is a negative regulator of *Mycobacterium tuberculosis* phagosome maturation in macrophages

Anetta Härtlova, Susanne Herbst, Julien Peltier, Angela Rodgers, Orsolya Bilkei-Gorzo, Antony Fearn, Brian D. Dill, Heyne Lee, Rowan Flynn, Sally A. Cowley, Paul Davies, Patrick A. Lewis, Ian G. Ganley, Jennifer Martinez, Dario R. Alessi, Alastair D. Reith, Matthias Trost and Maximiliano G. Gutierrez

Review timeline:

Submission date:	24 November 2017
Editorial Decision:	19 December 2017
Revision received:	15 February 2018
Editorial Decision:	23 March 2018
Revision received:	5 April 2018
Accepted:	24 April 2018

Editor: Hartmut Vodermaier

Transaction Report:

Please note that the manuscript was previously reviewed at another journal. Since the original reviews are not subject to The EMBO Journal's transparent review process policy, the reports and author response cannot be published.

1st Editorial Decision

19 December 2017

Thank you again for submitting your manuscript together with the previous reports and responses to The EMBO Journal for our consideration. I have now heard back from two arbitrating referees, who had agreed to look both at the paper as well as at the previous referees' concerns and your responses. In light of their comments, we shall be happy to consider a revised version further for publication, pending the addition of the revisions already proposed in your response letter, as well as addressing the various minor arbitrator comments and the first two major concerns of arbitrator 1. Among the points raised in the original review, assessing trafficking and bacterial growth in the presence of IFN gamma appears to be one of the key issues to be addressed during revision. On the other hand, in agreement with arbitrator 2 I conclude that further follow-up on the molecular mechanism of LRRK2 action would not be essential within the scope of the present study, nor would investigations of LRRK2 macrophage roles in conditional knock-outs be needed.

When preparing your revision and point-by-point response letter to the original and the new comments, please remember that our policy to allow only a single round of major revision makes it important to carefully answer to all points raised during this round. If necessary, we might also discuss an extension of the regular three-months revision period, during which any competing work published elsewhere would not negatively affect our final assessment of your own study.

Please also refer to the sections below for additional information on preparing, formatting and uploading a revised manuscript. Should you have any further questions related to this decision or your revision, please do not hesitate to contact me.

Thank you again for the opportunity to consider this study for The EMBO Journal! I look forward to receiving your revision.

REFEREE REPORTS

Arbitrator #1:

I have several major concerns:

The first one is that the effects they see with the LRRK2 KD or KO are quite small relative to the wild type. They actually used state of the art technique for subcellular trafficking, however I am cautious about their determination of intracellular CFUs by the old bacterial titration technique. Actually they see only a two-fold difference in numbers over a 3-days kinetic study (Fig 1A,B), which can be meaningful if at the same time a proper quantification of the macrophage survival is done on the same samples. This can easily be done by quantitative imaging and they authors should really provide this piece of data. Also standard controls should be used for the imaging of LAMP-1 recruitment, as for instance heat-killed bacteria; although we are confident that M. Gutierrez is really expert in this.

The second major concern is on the use of GSK2578215A inhibitor. This compound is used at quite high concentration- ie 1 microM- when specified. I could also not find the concentrations used in Figure 1D, are they higher? In log-2 scale? Similarly as above, as it is a compound targeting host signaling, could the authors check for the cytotoxicity of this compound on the macrophage. Indeed, at 30% toxicity in macrophages can already lead to a 50% loss of bacteria counts, which is the order of magnitude that they see. Also did the author show that in their conditions of Mtb-macrophage infection, the inhibitor actually impaired LRRK2 activity? This would be important to verify the target exposure.

The third major concern is about the molecular mechanism that involves LRRK2. As suggested by the reviewer, I believe more analysis should be done on this. The main question of "how does it affect phagosome maturation" is not answered.

The last point on in vivo is less critical. We all know that the mouse model does not reproduce human TB and to know more of the effect of LRRK2 in macrophages, I believe the author should resort to better tools available now, ie macrophage specific LRRK2 KO mice. One minor concern: Why are the elevated LC3B levels in LRRK2 KO macrophages (Fig. S2B-C) not detected in Fig. S2D before gamma interferon is added?

Arbitrator #2:

The study brings a substantial amount of novel knowledge on the role of the LRRK2 kinase in phagosome biology in general, and in *M. tuberculosis* intracellular trafficking and pathogenesis in particular.

The authors' reply to previous reviewers' comments are overall satisfactory. The link between in vitro and in vivo observations could be clarified by performing a limited number of additional experiments, in particular the one suggested by Referee 2, and could be better discussed.

Specific comments over previous reviews.

Reviewer 1

I agree with the authors that identifying the LRRK2 substrates, including Rab GTPases, and elucidating the very function of these substrates in the reported phenotypes would represent a formidable amount of additional work that goes far beyond the scope of the present manuscript. Other comments made by this Referee can be addressed in a revised version.

Reviewer 2

As proposed by the Referee, assessing the influence of IFN-gamma on bacterial growth in WT and KO macrophages is a key experiment that might help explain the phenotypes observed in vivo. Other comments made by this Referee can be addressed in a revised version.

Reviewer 3

I agree with the Referee that more work is needed to dissect the very mechanism(s) involved in LRRK2-mediated regulation of the PI3K complex and Rubicon. Yet, I consider that the present study, if properly revised to address the referees' comments, brings sufficient novel knowledge that will stimulate further research by the authors or other groups.

Additional comments

In Fig 3E, data using MLI-2 is not shown and in Fig 3F, data with WT only is not shown. This should be fixed.

Line 148 "(Figure 1N-O)" should be "(Figure 1N)".

Fig 5C in the text refers to panel D in the figure.

1st Revision - authors' response

15 February 2018

Many thanks for the reviews of our paper entitled "**LRRK2 is a negative regulator of *Mycobacterium tuberculosis* phagosome maturation in macrophages**". We have now addressed and elaborated on the reviewer's remarks.

Specifically, we have now added new data:

- Bafilomycin controls in WT and LRRK2 KO macrophages (new Fig. 2)
- A2016T KI controls in phagosomal proteolysis (new Fig. 3)
- Loading controls for phagosomal proteins (new Fig. 5)
- Additional siRNA for Rubicon (new Fig. 5)
- Effect of the inhibitor in macrophages (new Fig. S1)
- Cell viability assays (new Fig S1)
- Image-based analysis of survival (new Fig. S1)
- Gamma-interferon effect on survival (new Fig. S1)
- Total lysosomal morphology and activity (new Fig. S2)
- Autophagy flux controls (new Fig. S3)
- Sorting strategy for phagoFACS (new Fig. S4)
- Western blot for Rubicon KO and validation of antibody (new Fig. S5)

We believe that the new data and modified manuscript should satisfy the reviewer's and arbitrator's remarks and substantially reinforced the manuscript.

Arbitrator #1:

The first one is that the effects they see with the LRRK2 KD or KO are quite small relative to the wild type. They actually used state of the art technique for subcellular trafficking, however I am cautious about their determination of intracellular CFUs by the old bacterial titration technique. Actually, they see only a two-fold difference in numbers over a 3-days kinetic study (Fig 1A, B), which can be meaningful if at the same time a proper quantification of the macrophage survival is done on the same samples. This can easily be done by quantitative imaging and they should really provide this piece of data. Also, standard controls should be used for the imaging of LAMP-1 recruitment, as for instance heat-killed bacteria; although we are confident that M. Gutierrez is really expert in this.

We agree that quantitative imaging is sometimes more accurate than CFUs. We have included quantitative data showing bacterial replication and the results are similar (new Figure S1). We also

performed a control of the imaging with LAMP-1 comparing *Mtb* WT and the mutant RD1 (Lerner et al., 2017; Schnettger et al., 2017). We see a significant increase in the percentage of *Mtb* RD1 mutant positive for LAMP-1, as expected from our previous work (see figure below).

Lamp1 recruitment to Δ RD1 at 24 hrs p.i. (positive control):

The second major concern is on the use of GSK2578215A inhibitor. This compound is used at quite high concentration- ie 1 microM- when specified. I could also not find the concentrations used in Figure 1D, are they higher? In log-2 scale? Similarly, as above, as it is a compound targeting host signaling, could the authors check for the cytotoxicity of this compound on the macrophage. Indeed, at 30% toxicity in macrophages can already lead to a 50% loss of bacteria counts, which is the order of magnitude that they see. Also did the author show that in their conditions of *Mtb*-macrophage infection, the inhibitor actually impaired LRRK2 activity? This would be important to verify the target exposure.

This is a very important point, we have tested toxicity and it is not affected in macrophages (data is now included in new Figure S1). We would like to mention that this concentration is commonly used in most of the work using this inhibitor. Second point is also very important, it is generally difficult to have a marker of LRRK2 activation, so we have used the most common used one that measured the autophosphorylation in serine 935. We now provide this data (new Figure S1B).

The third major concern is about the molecular mechanism that involves LRRK2. As suggested by the reviewer, I believe more analysis should be done on this. The main question of "how does it affect phagosome maturation" is not answered.

Please see comments above.

The last point on in vivo is less critical. We all know that the mouse model does not reproduce human TB and to know more of the effect of LRRK2 in macrophages, I believe the author should resort to better tools available now, ie macrophage specific LRRK2 KO mice. One minor concern: Why are the elevated LC3B levels in LRRK2 KO macrophages (Fig. S2B-C) not detected in Fig. S2D before gamma interferon is added?

The arbitrator is right that the levels of LC3-II in LRRK2 KO macrophages seems not to be higher. However, we have done this analysis several times and this is the case (as reported in the literature and other figures of this work). In fact, if the LC3II/LC3-I ratio is considered, then it is higher in LRRK2 KO for that particular gel. We believe these differences could be attributed to different batches of the anti-LC3 antibody.

Arbitrator #2:

Reviewer 1: I agree with the authors that identifying the LRRK2 substrates, including Rab GTPases, and elucidating the very function of these substrates in the reported phenotypes would represent a formidable amount of additional work that goes far beyond the scope of the present manuscript. Other comments made by this Referee can be addressed in a revised version.

Thanks, we have addressed these comments.

Reviewer 2: As proposed by the Referee, assessing the influence of IFN-gamma on bacterial growth in WT and KO macrophages is a key experiment that might help explain the phenotypes observed in vivo. Other comments made by this Referee can be addressed in a revised version.

We agreed this was an important point. We have added these experiments (see above and new Figure S1), addressing the reviewer's comments.

Reviewer 3: I agree with the Referee that more work is needed to dissect the very mechanism(s) involved in LRRK2-mediated regulation of the PI3K complex and Rubicon. Yet, I consider that the present study, if properly revised to address the referees' comments, brings sufficient novel knowledge that will stimulate further research by the authors or other groups.

Thanks, we believe that is the revised version we addressed these concerns.

Additional comments

In Fig 3E, data using MLI-2 is not shown and in Fig 3F, data with WT only is not shown. This should be fixed. Line 148 "(Figure 1N-O)" should be "(Figure 1N)". Fig 5C in the text refers to panel D in the figure.

Many thanks. We now show the data with the WT condition (new Figure 3) and modified the text accordingly.

2nd Editorial Decision

23 March 2018

Thank you again for submitting your revised manuscript for our consideration. Our two arbitrating referees have now assessed it once more, and in light of their feedback copied below, we shall be happy to offer publication of the study, after some remaining editorial issues will have been addressed. As you will see, arbitrator 1 is concerned about the preparation of the revised version, including missing experimental detail, possibly discrepant figure legends, and insufficient discussion of newly added data - I therefore would like to ask you to address these issues in a final version, and to provide a response to these comments.

ARBITRATOR REPORTS

Arbitrator #1:

Concerning the first comment, the authors have included quantitative data from an image-based technique and displayed the results in new Figure S1. At present, the legend of Fig S1 does not match with the Figure display. Panels E, F, G, H and I should be exchanged by panels G, H, I, E and F. Also the authors refer only briefly to these data on line 150 mentioning that they use the same technique as in their Schnettger paper. I would have appreciated to have more details in the present manuscript.

Our second comment was on the putative cytotoxicity of GSK2578215A. The authors have monitored LDH release from Mtb infected macrophages in presence of the chemical inhibitor and the data are presented in Figure S1. Again, the legend of Fig S1 does not match with the Figure display. A, B and C should be C, A and B. Again, I could not find enough details about the experiment. There is no indication about the timing of the LDH measurement. Also it is strange that the authors have not reported LDH data for higher concentrations up to 10 microM, could this mean that the compound becomes toxic at 2 microM or higher. What is the selectivity ratio (activity/toxicity)?

I am not pleased about the "please see comments above" for my third concern. I agree that this could be the work of a new post-doc that will benefit from solid preliminary findings and substantial funding from the recently secured EU-ERC-CSG grant in the team. Currently the results lack the molecular mechanisms, which are now expected in the TB field for high impact papers. Indeed, there are more and more host proteins involved in Mtb phagosome maturation and there is hardly any detailed mechanistic study about them. The Gutierrez team is in a privileged situation to make real breakthrough in this field and the field really counts on them.

At this point, I wonder whether the authors have uploaded the right supplementary files. The authors discuss about the LCE3II/LC3-I ratio; however, on the Fig S3 there is no WB quantification. Overall, we feel that the revised version was written too hastily and would recommend the manuscript to be sent back to the authors.

Arbitrator #2:

The authors' revisions even further convinced me that this study brings a substantial amount of novel knowledge on the role of the LRRK2 kinase in phagosome biology in general, and in M. tuberculosis intracellular trafficking and pathogenesis in particular.

The authors' reply to previous reviewers' comments are now fully satisfactory, and the link between in vitro and in vivo observations has been clarified by performing additional experiments and is better discussed.

2nd Revision - authors' response

5 April 2018

Many thanks for the reviews and feedback of our paper entitled "**LRRK2 is a negative regulator of *Mycobacterium tuberculosis* phagosome maturation in macrophages**".

We apologise for the mislabelling of 2 panels in Figure S1, Figure S3 was actually okay. We have now corrected this. Moreover, we have also added more details on the methods used for image analysis as requested by the arbitrator 1. Please see attached point-by point reply.

Point by point reply to arbitrator 1:

Concerning the first comment, the authors have included quantitative data from an image-based technique and displayed the results in new Figure S1. At present, the legend of Fig S1 does not match with the Figure display. Panels E, F, G, H and I should be exchanged by panels G, H, I, E and F. Also the authors refer only briefly to these data on line 150 mentioning that they use the same technique as in their Schnettger paper. I would have appreciated to have more details in the present manuscript.

We sincerely apologize for the incorrect Fig S1 legend; it was due to a swap in the panels. We have now corrected this. We also added more information regarding the single cell quantification as described in Schnettger et al., 2017.

Our second comment was on the putative cytotoxicity of GSK2578215A. The authors have monitored LDH release from Mtb infected macrophages in presence of the chemical inhibitor and the data are presented in Figure S1. Again, the legend of Fig S1 does not match with the Figure display. A, B and C should be C, A and B. Again, I could not find enough details about the experiment. There is no indication about the timing of the LDH measurement. Also it is strange that the authors have not reported LDH data for higher concentrations up to 10 microM, could this mean that the compound becomes toxic at 2 microM or higher. What is the selectivity ratio (activity/toxicity)?

Again, apologies for the mislabeling of the panels in the Fig S1 that have been corrected now. We have now included a detailed protocol of the LDH measurements. We did not use concentrations higher than 1 μ M for long-term studies and therefore did not feel the need to test higher concentrations for toxicity. As reported by Reith et al, we confirmed that 1 μ M of GSK257821A achieves maximum LRRK2 kinase inhibition as measured by the abolished pS935 LRRK2 phosphorylation. Additionally, Reith et al reported an increase in off-target effects of the GSK257821A LRRK2 inhibitor at 10 μ M making it an unsuitable concentration to use.

Figure 1 GSK2578215A dose titration in BMDM WT macrophages. Data shows mean + SEM of three independent experiments.

I am not pleased about the "please see comments above" for my third concern. I agree that this could be the work of a new post-doc that will benefit from solid preliminary findings and substantial funding from the recently secured EU-ERC-CSG grant in the team. Currently the results lack the molecular mechanisms, which are now expected in the TB field for high impact papers. Indeed, there are more and more host proteins involved in Mtb phagosome maturation and there is hardly any detailed mechanistic study about them. The Gutierrez team is in a privileged situation to make real breakthrough in this field and the field really counts on them.

I understand the arbitrator's frustration. I believe this study does provide a mechanistic link between LRRK2 and PI3K activity via Rubicon on phagosome function. This aspect is very important for research on the LRRK2 in general and goes beyond tuberculosis. There has been great interest over the last decade in LRRK2 function and this study reports for first time a cellular function for LRRK2 in macrophages. This work is therefore not only important for the TB field but also for investigators working on Parkinson's disease, LRRK2 and innate immunity. I do agree that a mechanism linking the kinase activity of LRRK2 with phagosome function is crucial. However, working with this kinase is not an easy task. Only people working with this large kinase are aware of this and a key example is the fact that after 15 years of intensive research, the function of LRRK2 is still unknown. Moreover, we lack tools to investigate mechanisms in detail. It is likely that phosphorylation of Rab GTPases by LRRK2 contributes to the regulation of phagosome function. We are currently generating the tools to investigate this further. I hope our work will stimulate research on the cellular function of LRRK2 and its role in the innate immune response in general (as kindly suggested by arbitrator 2).

At this point, I wonder whether the authors have uploaded the right supplementary files. The authors discuss about the LC3II/LC3-I ratio; however, on the Fig S3 there is no WB quantification.

We did not refer to LC3II/LC3I ratio in the text or discussion. The WB quantification in Fig S3 was originally included and it shows the LC3II/tubulin ratio. The WB signal ratio between LC3-I and LC3-II has been used to determine changes in the extent of autophagy. However, due to differential levels of expression, a consensus has emerged whereby overall levels of LC3-II are normalized to a loading control, such as α -tubulin (Kimura et al., 2009).

3rd Editorial Decision

24 April 2018

Thank you for submitting your final revised manuscript for our consideration. After having assessed your final referee responses and carefully gone through all materials, I am pleased to inform you that we have now accepted it for publication in The EMBO Journal!

Corresponding Author Name: Maximiliano Gutierrez and Matthias Trost

Manuscript Number: EMBOJ-2017-98694R